# Incidence and Treatment Outcome of Rhinosinusitis before Kidney Transplantation: A Retrospective Cohort Study

**DOI:** 10.3390/jpm11060553

**Published:** 2021-06-14

**Authors:** Jin Seok Oh, Min Soo Kim, Sung Hee Kim, Ji Heui Kim

**Affiliations:** 1Department of Otorhinolaryngology—Head and Neck Surgery, National Medical Center, 245, Eulji-ro, Jung-gu, Seoul 04564, Korea; loveja01@nmc.or.kr (J.S.O.); minsonic@naver.com (M.S.K.); 2Department of Otorhinolaryngology—Head and Neck Surgery, Asan Medical Center, University of Ulsan College of Medicine, 88 Olympic-ro 43-gil, Songpa-gu, Seoul 05505, Korea

**Keywords:** immunodeficiency, sinusitis, endoscopy, imaging, computed tomography

## Abstract

Background: The use of immunosuppressants after transplantation can aggravate sinus infections. Although kidney transplantation (KT) recipients are administered strong immunosuppressant therapy, there is few consensus or reports on incidence and treatment of rhinosinusitis before KT. This study was undertaken to analyze the results of a cohort of KT recipients that underwent sinonasal evaluation before KT. Methods: Observational retrospective cohort data were analyzed from adults who underwent a KT between January 2015 and December 2018. In total, 966 patients were screened by clinical history, nasal endoscopy, and plain X-ray before KT. Results: A total of 86 patients (8.9%) were diagnosed with rhinosinusitis. Twenty-three of the eighty-six patients (26.7%) who underwent plain X-ray on second follow up were successfully treated with primary and secondary antibiotics, saline irrigation, and INS. From the remaining 63 patients who underwent additional CT on second follow up, 43 patients were treated with primary or secondary antibiotics and 20 patients (10 with chronic rhinosinusitis and 10 with fungal ball) were treated with endoscopic sinus surgery. There were no serious complications affecting patient mortality after KT. Conclusion: We report that 8.9% of patients showed abnormal findings in sinonasal evaluation before KT. Although most patients did not require surgery, surgery is recommended for active rhinosinusitis, which does not respond to medication, and for fungal rhinosinusitis to prevent postoperative sinonasal infection.

## 1. Introduction

Rhinosinusitis is a type of infection occurring in patients who undergo solid organ or stem cell transplantation (SCT). The incidence of rhinosinusitis in transplant patients varies with the type or organ of transplantation. In SCT, the incidence of rhinosinusitis before transplantation is approximately 10–16% [1,2], but it increases to as high as 44% during the post-transplantation period [3,4,5,6]. A recent review article advised that all patients should be screened using computed tomography (CT) before undergoing SCT, and those showing rhinosinusitis symptoms should be treated before transplantation.

However, there are limited data in the literature regarding rhinosinusitis in patients undergoing solid organ transplantation. One study demonstrated that the incidence of radiologically detected rhinosinusitis in patients before liver transplantation (LT) was 11.1% and suggested that untreated rhinosinusitis is correlated with increased mortality after LT. Therefore, experts recommend conducting a sinonasal evaluation before LT [7]. Our center reported that untreated rhinosinusitis often aggravated after LT [8].

Kidney transplantation (KT), a common type of solid organ transplantation, is performed for patients with end-stage renal disease (ESRD) requiring renal replacement therapy. These immunocompromised patients with ESRD are susceptible to infection. In particular, the use of immunosuppressants to decrease the risk of rejection after KT can increase patients’ vulnerability to various infections. Although KT recipients are administered stronger immunosuppressant therapy than LT recipients [9,10], there are few objective data for the screening and management of rhinosinusitis in immunocompromised KT recipients.

Our center routinely performs sinonasal evaluation before KT regardless of the patient’s sinonasal symptoms. Here, we analyzed the outcomes of a cohort who underwent sinonasal evaluation before KT and suggest treatment guidelines for rhinosinusitis prior to KT.

## 2. Materials and Methods

### 2.1. Study Population

This retrospective cohort study was approved by the institutional review board of the Asan Medical Center (2016-0929) in Seoul, South Korea.

We analyzed all observational retrospective cohort data from 1111 adult patients who underwent KT at the Asan Medical Center between January 2015 and December 2018. Of the 1111 KT recipients, 966 (87.0%) were examined by an otorhinolaryngologist before KT using nasal endoscopy and plain X-ray, including Water’s, Caldwell’s, and the lateral view, and analyzing clinical history for the existence of an infection source in the sinonasal cavity. The remaining 145 patients (13.0%) underwent KT without screening by the otorhinolaryngologist due to an urgent KT schedule.

### 2.2. Protocol and Procedure for Sinonasal Evaluation and Treatment

Patients exhibiting active inflammation, such as endoscopic signs of nasal polyps, purulent discharge, and/or edema, at initial examination, were prescribed amoxicillin–clavulanate acid at 625 mg, three times a day for 1–2 weeks, combined with saline irrigation and intranasal steroid (INS) to eliminate and treat bacterial rhinosinusitis. These patients were re-examined using nasal endoscopy and plain X-ray or additional CT. Patients with persistent symptoms and signs of chronic rhinosinusitis (CRS) despite sufficient medication combined with saline irrigation and INS or those with suspected fungal rhinosinusitis on CT underwent endoscopic sinus surgery (ESS). Acute rhinosinusitis (ARS) and CRS were diagnosed according to the guidelines of the American Academy of Otolaryngology–Head and Neck Surgery Foundation for adult rhinosinusitis [11]. Fungal rhinosinusitis was confirmed by intraoperative findings and biopsy. Demographic data, Lund–Kennedy (LK) endoscopic scores [12], Lund–Mackay (LM) CT scores [13], treatment modality, and treatment outcomes were obtained from medical records and imaging findings.

### 2.3. Immunosuppressant Therapy for KT Recipients

For maintenance immunosuppressant therapy for the KT recipients, a triple regimen of calcineurin inhibitors (tacrolimus or cyclosporine) plus mycophenolate mofetil or azathioprine plus glucocorticoids was used [8].

### 2.4. Statistical Analyses

The Kruskal–Wallis test was used to determine the statistically significant differences in LK and LM scores between the CRS groups. All data were analyzed using IBM^®^ SPSS^®^ Statistics for Windows, version 20.0 (SPSS Inc., Chicago, IL, USA). A *p* value of <0.05 was considered statistically significant.

## 3. Results

Among the 966 patients evaluated by the otorhinolaryngologist before KT, 86 (8.9%) were diagnosed with rhinosinusitis: 23 (26.7%) with ARS, 53 (61.6%) with CRS, and 10 (11.7%) with fungal ball. Approximately two-thirds of the patients were male (59 male, 27 female), and the median patient age was 32.3 (range, 21–60) years. Nasal polyps (12.8%), purulence (19.8%), and/or middle meatal edema (40.7%) were observed on nasal endoscopic examination, and the mean LK score was 1.5 (range, 1–10).

Overall, patients with rhinosinusitis before KT were treated with primary antibiotics, nasal saline irrigation, and INS, and underwent a secondary follow-up in 2 weeks. Of the 86 patients, 23 (26.7%) had rhinosinusitis symptoms <3 months and underwent plain X-ray on second follow-up; these patients were successfully treated with primary and secondary antibiotics, saline irrigation, and INS (Figure 1). They were diagnosed with ARS.

Of the 86 patients, 63 (73.3%) who had rhinosinusitis symptoms for >3 months underwent CT scan after primary therapy, and 53 (61.6%) patients showed mucosal thickening on CT scan. Further, of the 53 patients, 27 (50.9%) who did not show residual symptoms and purulent discharge after primary management underwent KT without any further management; 16 (30.2%) underwent KT after sufficient treatment with secondary antibiotics, saline irrigation, and INS; and the remaining 10 (18.9%) who had active symptoms and endoscopic findings despite secondary management underwent ESS before KT. In addition, 10 (11.7%) of the 86 patients who underwent nasal endoscopy and CT were suspected to have noninvasive fungal rhinosinusitis on CT scans and underwent KT after being successfully treated with ESS. Finally, none of the patients evaluated by the otorhinolaryngologist before KT developed septic conditions or systemic fungal infections due to rhinosinusitis.

When the LK and LM scores of the patients with CRS were compared in terms of the treatment modality, the mean scores of the patients treated with ESS were significantly higher than those of patients treated with primary and secondary antibiotics (*P* < 0.001 and *P* = 0.009, respectively; Figure 2). The mean postoperative antibiotic treatment duration was 2.05 (range, 0–4) weeks. Furthermore, the mean period of complete healing was 3.65 (range, 2–28.4, 95% confidence interval [CI] 2.84–9.39) months. 

Among the 880 patients without evidence of rhinosinusitis before KT, 12 (1.4%) developed rhinosinusitis symptoms and signs after KT and were treated successfully with medication (*n* = 7) or ESS (*n* = 5). There were no local complications or sepsis due to rhinosinusitis.

## 4. Discussion

Patients with ESRD are immunocompromised because of the illness itself, and they become more susceptible to infections due to immunosuppressant administration after KT. Although rhinosinusitis is one of the most common infectious diseases in the population, there are few reports on its incidence in patients with ESRD before KT and its evaluation and management protocols. In our current study, we report the incidence of rhinosinusitis before KT as 8.9%. Although most of the patients did not require surgery, surgery is recommended for active rhinosinusitis that does not respond to medication and for fungal rhinosinusitis to prevent postoperative sinonasal infection.

Previous studies have typically focused on the sinonasal evaluation of SCT recipients before the initiation of immunosuppressants [6,14,15]. Billings et al. reported that the severity of radiographic rhinosinusitis on pre-SCT CT scans was correlated with post-SCT clinical and radiographic rhinosinusitis and was also associated with a trend of decreased survival [15]. Another article suggests screening all patients using CT before undergoing SCT and treating those with rhinosinusitis symptoms before transplantation. To minimize the local as well as generalized spread of infection that may prove to be fatal, it is safer for patients to assume that they have rhinosinusitis and to undergo treatment rather than neglecting this possibility [6].

In cases of solid organ transplantation, pre-transplantation sinonasal workup has rarely been discussed in the literature, and studies differ depending on the solid organ being transplanted. In LT recipients, the overall rate of infectious rhinosinusitis was 7.3%, and untreated rhinosinusitis may increase infection-related mortality after LT. Thus, sinonasal evaluation before LT is recommended [7]. In contrast, in KT recipients, the prevalence of pretransplant rhinosinusitis was 4.2%, and the prevalence and recurrence rate of rhinosinusitis did not increase after KT. Therefore, routine sinonasal evaluation for asymptomatic patients before KT is not recommended [16].

However, in this study, the overall detection rate of rhinosinusitis before KT was 8.9%, which was not low; this necessitated medical or surgical treatment before KT. In addition, septic conditions or systemic fungal infections caused by rhinosinusitis did not occur in any of the patients who underwent sinonasal evaluation before KT. In our center, patients with abnormal findings at initial screening are prescribed primary antibiotics, nasal saline irrigation, and INS to eliminate and treat bacterial rhinosinusitis. Through this approach, secondary antibiotics or surgery was not needed in 43 (50.0%; 16 and 27 patients with ARS and CRS, respectively) of the 86 patients with this condition. In addition, the LK and LM scores of the surgically treated patients with CRS were significantly higher, suggesting that mild active rhinosinusitis can be treated with antibiotics in combination with saline irrigation and INS and that surgical treatment can be minimized. Their postoperative antibiotic treatment and time needed for complete healing appeared similar to those of patients without ESRD, indicating that surgery should be actively considered if rhinosinusitis is not treated using secondary antibiotics, nasal saline irrigation, and INS. There were no serious complications affecting patient mortality after KT, probably because most patients were evaluated routinely by an otorhinolaryngologist and active rhinosinusitis was treated completely with medication or surgery.

The overall prevalence of CRS in Korea is 6.95% [17]. Among sociodemographic factors, male sex, old age, and high stress are significantly related to CRS in the Korean population. The prevalence of CRS was reported to be higher in males (8.24%) than in females (6.0%) and the lowest (3.95%) in younger adults (aged 19–29 years) and the highest (10.16%) in elderly patients (age > 70 years) [17,18]. In this study, we found that the prevalence of CRS in pre-KT patients was 5.49% (53/966), which is slightly lower than the overall prevalence of CRS in Korea. 

Unfortunately, the impact of this study is limited by its retrospective design. In addition, there are no data on how the group treated for rhinosinusitis before transplantation affected the clinical outcome after transplantation. However, most transplant centers currently perform an otolaryngological examination before KT, and patients who need treatment undergo transplantation after treatment. Therefore, in this study, we place more focus on the guidelines for the screening and management of rhinosinusitis in immunocompromised patients among solid organ transplantation recipients [19].

Considering the cost and time involved, the need for sinonasal evaluation in asymptomatic solid organ transplantation recipients is debatable, especially for KT recipients. In the case of deceased donor KT, the risk of infection after transplantation is higher due to the lack of time for pre-transplantation workup. Therefore, in future studies, it might be helpful to clarify the necessity of sinonasal evaluation before KT by comparing the sinonasal infections in the post-transplantation periods between KT from living and deceased donors.

In the current study, approximately 8.9% of the patients with ESRD showed abnormal findings in the sinonasal evaluation before KT. Although most patients did not require surgery, surgery is recommended for those with active rhinosinusitis that does not respond to medication and for patients with fungal rhinosinusitis to prevent postoperative sinonasal infection.

## Figures and Tables

**Figure 1 jpm-11-00553-f001:**
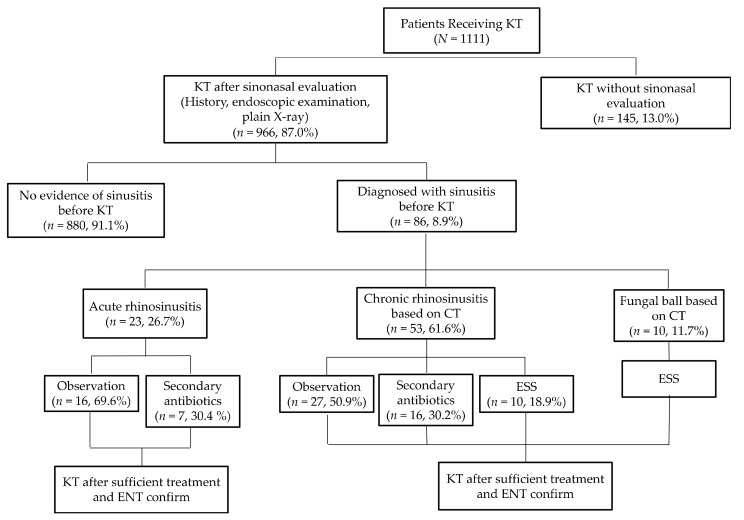
Flowchart of patients who underwent a sinonasal evaluation before kidney transplantation. KT, kidney transplantation; ENT, ear, nose, and throat specialist (otorhinolaryngologist); OMU CT, ostiomeatal unit computed tomography; ESS, endoscopic sinus surgery.

**Figure 2 jpm-11-00553-f002:**
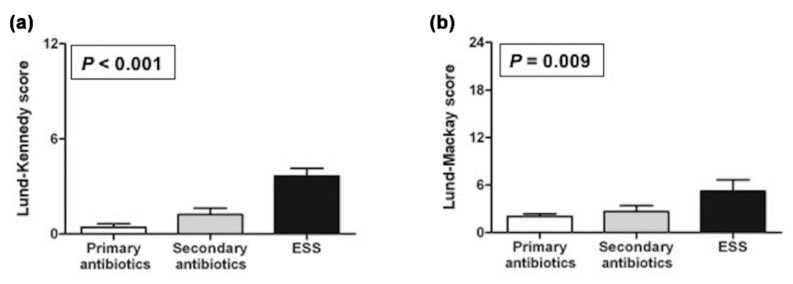
Lund–Kennedy scores of endoscopic assessment (**a**) and Lund–Mackay scores of computed tomography (**b**) in 53 patients with chronic rhinosinusitis before kidney transplantation. Both scores were significantly higher in patients treated with endoscopic sinus surgery (ESS) (*n* = 10) than in those treated with primary (*n* = 27) or secondary (*n* = 16) antibiotics.

## Data Availability

The data presented in this study are available on request from the corresponding author. The data are not publicly available due to ethical concerns.

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
