# Peer review of "Incidence and Treatment Outcome of Rhinosinusitis before Kidney Transplantation: A Retrospective Cohort Study"

_jpm, 2021, doi:10.3390/jpm11060553_

Round 1

Reviewer 1 Report

Dear Authors,

It was an interesting study which analyzed the relationship between Kidney transplantation and sinusitis.

The concerns of this study raised and should be addressed were;

1) It was not clear why this study included the data with the KT patients without sinonasal evaluation (n=145). If this data was excluded, then the prevalence of post-KT (not including the data who could have had sinusitis before KT) would be clear. 

2) Figure 1 looks a bit confusing.  The sentence at line 114 to 116 should be placed in the explanation of Figure 1.?

3) Is there any citation for "the guidelines for the screening and management of sinusitis in immunocompromised patients in the solid organ transplantation recipient group" (Line 191-192)

Reviewer 2 Report

Line 114-116: Looks like this should have been a figure caption?

Line 134: If the range is 2-28 months but the mean is 3.6, I would recommend to include confidence intervals.

Line 162: typing error?

Line 164: typing error!

Figure 1: please supply n

Discussion:

  • please make a statement and supply (if there is any) data about possible regional characteristics of the incidence of sinusitis in Korea
  • in my opinion, one of the most interesting questions is about the subgroups of symptomatic / asymptomatic patients. Thus I would recommend to include this subgroups both in analysis and discussion 
